# Priming Effects of Water Immersion on Paired Associative Stimulation-Induced Neural Plasticity in the Primary Motor Cortex

**DOI:** 10.3390/ijerph17010215

**Published:** 2019-12-27

**Authors:** Daisuke Sato, Koya Yamashiro, Yudai Yamazaki, Koyuki Ikarashi, Hideaki Onishi, Yasuhiro Baba, Atsuo Maruyama

**Affiliations:** 1Department of Health and Sports, Niigata University of Health and Welfare, Shimamicho 1398, Kita-ku, Niigata City, Niigata 950-3198, Japan; yamashiro@nuhw.ac.jp (K.Y.); baba@nuhw.ac.jp (Y.B.); 2Institute for Human Movement and Medical Science, Niigata University of Health and Welfare, Shimamicho 1398, Kita-ku, Niigata City, Niigata 950-3198, Japan; hwd17010@nuhw.ac.jp (Y.Y.); wtm19001@nuhw.ac.jp (K.I.); onishi@nuhw.ac.jp (H.O.); 3Graduate School, Niigata University of Health and Welfare, Shimamicho 1398, Kita-ku, Niigata City, Niigata 950-3198, Japan; 4Department of Rehabilitation Medicine, Kagoshima University, Sakuragaoka 8-35-1, Kagoshima City, Kagoshima 890-8520, Japan; atu-maru@m.kufm.kagoshima-u.ac.jp

**Keywords:** water immersion, m1 plasticity, pas25, short latency afferent inhibition

## Abstract

We aimed to verify whether indirect-wave (I-wave) recruitment and cortical inhibition can regulate or predict the plastic response to paired associative stimulation with an inter-stimulus interval of 25 ms (PAS25), and also whether water immersion (WI) can facilitate the subsequent PAS25-induced plasticity. To address the first question, we applied transcranial magnetic stimulation (TMS) to the M1 hand area, while alternating the direction of the induced current between posterior-to-anterior and anterior-to-posterior to activate two independent synaptic inputs to the corticospinal neurons. Moreover, we used a paired stimulation paradigm to evaluate the short-latency afferent inhibition (SAI) and short-interval intracortical inhibition (SICI). To address the second question, we examined the motor evoked potential (MEP) amplitudes before and after PAS25, with and without WI, and used the SAI, SICI, and MEP recruitment curves to determine the mechanism underlying priming by WI on PAS25. We demonstrated that SAI, with an inter-stimulus interval of 25 ms, might serve as a predictor of the response to PAS25, whereas I-wave recruitment evaluated by the MEP latency difference was not predictive of the PAS25 response, and found that 15 min WI prior to PAS25 facilitated long-term potentiation (LTP)-like plasticity due to a homeostatic increase in cholinergic activity.

## 1. Introduction

The use-dependent changes in synaptic strength (plasticity) are essential for learning and memory [1]. Among the multiple types of synaptic plasticity described thus far [2,3], Hebbian long-term potentiation (LTP) and depression (LTD) are the most thoroughly studied as potential neuroplastic mechanisms for learning. LTP and LTD are mutually dependent as induction thresholds, and are influenced by the prior history of neuronal activity and functional state of the synapse, termed as metaplasticity [4]. Metaplasticity represents a neuroprotective mechanism that stabilizes synaptic weights in neuronal networks while maintaining the capacity for synaptic plasticity by varying the induction thresholds according to the integrated postsynaptic activity [5]. 

Various non-invasive transcranial magnetic stimulation (TMS) protocols can be used to explore the neurophysiological mechanisms underlying synaptic plasticity in the human cortex. In particular, paired associative stimulation (PAS) [6] involves the application of an electrical stimulus to the median nerve at the wrist, followed by TMS over the primary motor cortex (M1) after 25 ms (PAS25). Repeated pairings at this interval increase the corticospinal excitability for 30–60 min, which is measured as motor-evoked potentials (MEPs). However, recent studies show that PAS25-induced plasticity is not observed in 25%−64% of healthy young volunteers [7,8,9,10,11]. Numerous factors have been proposed to explain this non-responsive population, such as in the review article by Ridding and Ziemann [12]. As non-responsiveness affects both research design and the extent of potential therapeutic applications, we investigated the factors that may predict or alter the plastic response to PAS25.

A recent study reported that the individual response variability to theta burst stimulation (TBS) was strongly influenced by the recruited interneuron networks [13]. Thus, we first questioned whether an analogous recruitment process accounts for individual variability in the PAS25 response; in particular, we examined whether the variation in potentiating indirect-wave (I-wave) recruitment influences the PAS25-induced plasticity. Furthermore, γ-aminobutyric acid (GABA)ergic interneurons have been reported to modulate the plastic response to PAS25 [14,15]. In fact, cortical GABAergic activity negatively interacts with short-latency afferent inhibition (SAI) [16,17]. As the PAS25 protocol is actually a repeated SAI protocol, it is likely that PAS25-induced potentiation may be associated with a progressively decreasing inhibition by the conditioning afferent stimuli on later I-wave recruitment [15]. Thus, we predicted that individual SAI with an inter-stimulus interval (ISI) of 25 ms may also be associated with the plastic response to PAS25, because both SAI with an ISI of 25 ms and PAS25 recruit the same sensorimotor networks. 

The history of prior cortical activity may be another possible factor influencing the PAS25 response [8,18,19]. Water immersion (WI) is one simple way of modulating the cortical activity in the sensorimotor system [20,21]. Our previous study showed that WI decreased SAI [21] through a form of cortical inhibition that originates from the central cholinergic modulation of inhibitory circuits different from those underlying short-interval intracortical inhibition (SICI) [22]. Thus, WI may suppress the specific inhibitory circuits in the M1 that are modulated by cholinergic activity. However, effects of WI on cholinergic activity have remained unclear. Considering the characteristics of cholinergic activity, which contribute to the homeostatic modulation of sleep and wake states [23], cholinergic activity may invert after WI due to a homeostatic response. In that case, WI would promote PAS25 because the effects of PAS25 could be strengthened by cholinergic activity [24].

Here, we tested whether WI primed PAS25 by altering the cholinergic modulation of inhibitory circuits and M1 excitability. The elucidation of the effects of WI priming on the neural plasticity of M1 could increase the reliability of PAS25 induction for research on human cortical sensorimotor integration and synaptic plasticity, and could facilitate the development of improved aquatic therapies for neurological patients. 

## 2. Materials and Methods

### 2.1. Subjects

A total of 27 right-handed subjects (defined by the Edinburgh handedness questionnaire; Oldfield, 1971) participated in this study. We enrolled 18 participants for experiments 1 and 2 and nine for experiment 3. Informed consent was obtained from all subjects. The present study was conducted in accordance with the Declaration of Helsinki and approved by the ethics committee of Niigata University of Health and Welfare (approval number 17656-160404). 

### 2.2. TMS

TMS was performed by using two Magstim 200 stimulators (Magstim, Dyfed, United Kingdom) connected via a Y-cable to a figure-of-eight-shaped coil, with an internal wing diameter of 7 cm. The coil was held with the handle pointing backward and 45° laterally to the interhemispheric line to induce an anteriorly directed current in the brain, and was optimally positioned to obtain MEPs in the abductor pollicis brevis (APB) muscle. We verified the target position of the coil relative to the brain anatomy by using a frameless TMS navigation system (Brainsight, Rogue Resolution, United Kingdom). The position of the coil was fixed using this system throughout all the experiments. 

The resting motor threshold (RMT) was defined as the minimum stimulation intensity over the motor hotspot required to evoke an MEP of ≥50 μV in 5 out of 10 trials [25]. The active motor threshold (AMT) was defined as the lowest intensity required to evoke an MEP of 200 μV in more than 5 of 10 consecutive trials, whereas the subjects maintained approximately 10% contraction of the target muscle. To maintain 10% contraction, as performed by previous studies [26], a rectified running average EMG with an averaging window of 175 ms was used to provide visual feedback on the monitor to the participants.

### 2.3. Electromyographic Recording

Surface electromyographic (EMG) recordings were acquired in a belly-to-tendon montage from the APB and the first dorsal interosseous (FDI) muscle of the right hand. The raw signal was amplified and band-pass filtered between 5 Hz and 2 kHz (AB-601G, Nihon Kohden, Tokyo, Japan), and transferred through a Micro 1401 Laboratory Interface (Cambridge Electronic Design, Cambridge, United Kingdom) to a personal computer for further offline analysis. All electrodes were covered with a transparent film (Tegaderm Hydrocolloid Dressing, 3M Japan, Tokyo, Japan) for waterproofing, as described previously [27]. The waterproofing was applied before starting the experiment and removed after the final assessment in all experiments.

### 2.4. PAS Session

The PAS session comprised the application 180 electrical stimuli to the right median nerve at the wrist, paired with a single posterior-to-anterior (PA)-directed TMS over the hotspot of the left APB muscle. Square-wave pulses (duration, 200 µs) at three times the sensory threshold were applied through a bipolar electrode (cathodal proximal). TMS was delivered through a figure-of-eight-shaped coil connected to a Magstim 200 magnetic stimulator that was held in the same position, by using the brain navigation system described above. Stimulus pairs were delivered at 0.2 Hz.

Stimulation was applied at an intensity adjusted to evoke an MEP in the relaxed APB muscle of approximately 1 mV peak-to-peak at baseline (TS_1mV_base_). The effects of PAS with an ISI of 25 ms between the peripheral and TMS stimulus (PAS25) on the MEP amplitude were tested. This paradigm has been shown to induce a long-lasting MEP increase [6,28]. Subjects were instructed to look at their stimulated hand and count the peripheral electrical stimuli they perceived; they were then asked the actual count by the investigator three or four times during the PAS session [28]. The MEPs evoked in the APB were stored for off-line analysis. 

### 2.5. WI Intervention

The WI intervention lasted for 15 min. The subjects wore only swimwear and were seated on a comfortable reclining armchair with a mounted headrest. The subjects assumed the same body position under all interventions, and three belts were secured around the thigh, abdomen, and chest to prevent movement. Both the right and left hands were also fixed in the same relaxed position on the armrests by using a belt to prevent muscle contractions in water. We monitored the EMG for any contraction in the APB and FDI muscle during WI. Participants were instructed to focus their gaze at the wall facing them throughout the experiments to divert their attention from their right hand. For each intervention, the ambient air temperature was maintained at 30 ± 1 °C, and water temperature was maintained at 34 ± 1 °C. The ambient air and water temperatures were controlled to minimize the effect of changing skin temperature. The tank in which the subjects were seated was filled with water up to the axillary level. 

### 2.6. Experimental Design and Parameters

Figure 1 shows the overall design of experiments 1, 2, and 3, including all the interventions and parameters. All trial blocks were conducted in random order for each subject, with an interval of at least 1 week. All experiments were performed in the afternoon to control for circadian influences on motor excitability and plasticity [29]. 

#### 2.6.1. Experiment 1

First, RMT and AMTs with posterior-to-anterior (PA), anterior-to-posterior (AP), and lateral-to-medial (LM) currents—defined as AMT_pa_, AMT_ap_, and AMT_lm_—were measured; thereafter, MEPs during contraction were measured to evaluate the MEP onset latencies. All measurements were performed at the hotspot determined for PA currents, as previous experiments have shown that the direction of the current does not significantly influence the position of the hotspot [30,31].

We measured the onset latency of MEPs using the PA, AP, and LM currents during mild contractions (<10% of the maximum voluntary contraction) of the target muscle (Figure 2), according to the method of Hamada, Murase, Hasan, Balaratnam, and Rothwell [13]. Previous studies found that PA currents preferentially activate early I-waves, whereas AP currents mainly activate later I-waves (e.g., I3) [31,32,33]. Therefore, we decided to use the latency difference between the LM- and AP-evoked MEP onset as a measure of late I-wave recruitment (the longer the latency difference, the more efficient the later I-wave recruitment). 

M1 excitability was assessed by measuring the peak-to-peak MEP amplitude from the relaxed right APB muscle in response to a single-pulse TMS with PA current over the contralateral APB hotspot. The test stimulus (TS) intensity was fixed to produce MEPs of approximately 1.0 mV in the right APB muscle at baseline (TS_1mV_base_). The TS was applied every 5 s. During the experiments, EMG activity of the APB muscle was monitored using an oscilloscope. Trials contaminated with voluntary EMG activity were discarded from the analysis. M1 excitability assessments consisted of baseline measurements, one measurement between the priming session and PAS25 session (“between”), and three “post” measurements after the PAS25 session: immediately, 15 min, and 30 min after PAS25 (post0, post15, and post30, respectively). 

#### 2.6.2. Experiment 2

Paired TMS pulses were administered through the same stimulating coil over the left motor cortex to assess SICI. The test stimulus intensity was adjusted to elicit an unconditioned test MEP in the relaxed left APB of approximately 1 mV peak-to-peak amplitude (TS_1mV_adjusted_). Test stimuli were applied every 5 s. SICI was tested with a subthreshold conditioning stimulus (CS_SICI_) at 90% of the AMT, applied 3 ms before the TS, as described in previous studies [34,35]. 

SAI was studied by using a previously described protocol [36,37,38]. Conditioning electrical pulses (CS_SAI_ duration, 200 µs) were applied through a bipolar electrode to the right median nerve at the wrist (cathode proximal). The intensity of the conditioning stimulus was set at approximately three times the sensory threshold. The intensity of the TMS test pulse over the left motor cortex was adjusted to evoke an unconditioned MEP in the relaxed APB of approximately 1 mV peak-to-peak. The conditioning stimulus to the median nerve preceded the TMS test pulse by inter-stimulus intervals (ISIs) set according to the individual latency of the median nerve somatosensory evoked potential N20 component. To record the somatosensory evoked potentials, the active electrode was attached at a site 3 cm posterior to C3 (according to the international 10–20 system), and the reference was set at Fz. A total of 500 responses were averaged to identify the latency of the N20 peak. The ISIs corresponding to the N20 latency plus 2 and 4 ms, and an ISI of 25 ms were investigated [36]. Test stimuli were applied every 5 s. During the experiments, the EMG activity of the APB muscle was monitored using an oscilloscope. Trials contaminated with voluntary EMG activity were discarded.

After SEP measurements, RMT, AMT, TS_1mV_base_, and CS_SICI_ were established, and the baseline measurement block was started. Each measurement block consisted of five different stimuli: test alone, SICI with an ISI of 3 ms, SAI with an ISI of N20 latency plus 2 or 4 ms, and an ISI of 25 ms. The order was randomized by a computer, and 12 trials of each type were recorded per block. All assessments consisted of baseline measurements and six post measurements after intervention (non-WI and WI). Four post measurements were acquired after intervention with a short break of 1.5 min (post0, post1, post2, and post3), and two additional post measurements were acquired 15 and 30 min after post3 (post4, post5). The post1 and -2 measurements were acquired to examine the cholinergic and GABAergic neural activity, respectively, during the PAS25 session in experiment 1, because cholinergic activity has been shown to enhance PAS25-induced plasticity [24]. However, there is no evidence for changes in GABAergic and cholinergic neuronal activity immediately after WI [39]. 

During the experiments, the EMG activity of the APB muscle was monitored using an oscilloscope. Trials contaminated with voluntary EMG activities were discarded. The mean amplitudes of the conditioned MEPs in response to these stimulus paradigms were expressed as a percentage of the mean amplitude of the corresponding unconditioned test MEP. These data were averaged across all stimulus paradigms to obtain a grand mean single value of SICI_3ms_, SAI_N20+2_, SAI_N20+4_, and SAI_25ms_.

#### 2.6.3. Experiment 3

After evaluating RMT, we measured the MEPs evoked by single TMS pulses of increasing stimulus intensity (50%, 80%, 90%, 100%, 110%, 120%, 130%, and 150% of the RMT) in each participant to construct individual MEP recruitment curves. A total of eight pulses were delivered for each stimulus intensity in random order. Stimuli were delivered every 5 s. To avoid startle and reflex responses, we excluded the first MEP for each trial from the analysis. The assessment consisted of baseline measurements during WI and five post-WI measurements—three starting 4 min after WI (post0, post1, and post2) and two additional post assessments 15 and 30 min after post2 (post3 and post4). 

### 2.7. Data Analysis and Statistics

The participant characteristics and both TMS and electrical stimulation (ES) parameters for experiments 1, 2, and 3 are presented in Table 1. The intraclass correlation coefficients (ICCs) were calculated to test the consistency of the latency values in experiment 1 and the inhibited parameters in experiment 2 between the no priming and priming WI trials. 

#### 2.7.1. Experiment 1

MEP amplitudes were entered into a two-way repeated-measures ANOVA (rmANOVA) with “trial” (no priming and priming WI) and “time” (baseline, between, post0, post15, and post30) as within-subject factors. The MEP amplitudes normalized to baseline were entered into an rmANOVA, with “trial” (no priming and priming WI) and “time” (baseline, between, post0, post15, and post30) as within-subject factors. 

The PAS25 effects were also assessed by the grand averaging of the normalized MEP amplitudes measured at post0–post30, to evaluate the correlations between all the measurement values and the PAS25 effects. Pearson’s correlation coefficients were calculated to measure the strengths of these correlations. Moreover, the responder and non-responder groups were defined operationally according to the presence of grand average PAS25 responses of below and above 1. We primarily aimed to elucidate factors that were predictive of or that regulated M1 plasticity in response to PAS25, and hence, we performed rmANOVA with “trial” and “time” as within-subject factors separately in responders and non-responders. 

#### 2.7.2. Experiment 2

SICI_3ms_, SAI_N20+2_, SAI_N20+4_, and SAI_25ms_ were entered into a two-way rmANOVA, with “trial” (CON and WI) and “time” (baseline, post0, post1, post2, post3, post4, and post5) as within-subject factors. We calculated the Pearson’s correlation coefficients between the grand average normalized MEP amplitudes measured at post1−3 in experiment 1 and each inhibitory measure in experiment 2, in order to evaluate the strengths of the associations with PAS25. Because each experiment was conducted on a separate day, the ICCs were calculated to test the consistency of the inhibitory measures in experiment 2 between the CON and WI trials.

#### 2.7.3. Experiment 3

The MEP amplitudes were analyzed by two-way rmANOVA, with “intensity” (from 50% to 150% RMT) and “time” (baseline, during, post0, post1, post2, post3, and post4) as the within-subject factors. 

In all analyses using rmANOVA, the Greenhouse–Geisser correction was used, if necessary, to correct for non-sphericity, whereas Tukey’s post hoc tests were used for pair-wise comparisons. A *p*-value of <0.05 was considered significant. Data were analyzed using the Statistical Software Package (IBM SPSS Version 18, USA). All data are expressed as mean ± standard error of the mean (SEM). 

## 3. Results

All 18 participants completed 2 sessions in experiments 1 and 2, and 9 completed Experiment 3. 

### 3.1. Latency Difference among Different Coil Orientations

As explained in the Methods section, D-wave latency was estimated following the onset of large MEPs evoked by LM stimulation. We estimated I-wave circuit activation during PAS25 by measuring the onset latency of the near-threshold MEPs evoked by AP stimulation, relative to D-wave activation. Finally, we also measured the onset latencies of the near-threshold MEPs evoked by PA stimulation. Table 1 shows the MEP latencies for each TMS stimulus direction and the latency differences. 

The interclass correlation coefficients (ICC) calculated to test consistency between latency values measured in the same individual on different days (no priming and priming WI, Figure 3) were 0.921 for PA latency, 0.926 for AP latency, 0.905 for LM latency, 0.852 for the PA–LM latency difference, and 0.874 for the AP–LM latency difference (all *p* < 0.001), indicating that the data spread was almost entirely due to inter-individual differences in the TMS response. 

### 3.2. PAS25-Induced Plasticity in M1

Figure 3 shows the time course of PAS25-induced plasticity in the no priming trials (Figure 4A) and priming WI trials (Figure 4B) for all 18 participants, as well as the group average in both the trials (Figure 4C). 

Two-way rmANOVA of the MEP amplitudes revealed a significant interaction between “trial” and “time” (*F* [4, 68] = 9.417, *p* < 0.01), as well as the main effects of both “trial” (*F* [1, 17] = 13.539, *p* < 0.01) and “time” (*F* [4, 68] = 7.256, *p* < 0.01). Post hoc comparisons revealed significant differences in the values at post15 and post30, compared to those at baseline and the “between” time point only in the WI priming trials (*p* < 0.05, Bonferroni-corrected). Moreover, there were significant differences in the values between the WI and non-WI (CON) trials at post0, post15, and post30 (*p* < 0.05, Bonferroni-corrected).

Two-way rmANOVA of the MEP amplitudes normalized to baseline revealed a significant interaction between “trial” and “time” (*F* [4, 68] = 10.784, *p* < 0.01), as well as the main effects of “trial” (*F* [1, 17] = 62.882, *p* < 0.01) and “time” (*F* [4, 68] = 6.635, *p* < 0.01). Post hoc comparisons indicated significant differences in the values at post15 and post30, compared to those at baseline and the “between” time point only in the WI priming trials (*p* < 0.05, Bonferroni-corrected). There were also significant differences between no priming and WI priming trials at post0, post15, and post30 (*p* < 0.05, Bonferroni-corrected).

Figure 4D,E shows the group average time courses of PAS25-induced plasticity for responders and non-responders. Two-way rmANOVA of MEP amplitudes normalized to baseline revealed a significant interaction between “trial” and “time” in both responders (*F* [4, 36] = 5.059, *p* < 0.01) and non-responders (*F* [4, 28] = 6.672, *p* < 0.01). Moreover, the main effects of “trial” and “time” in responders (*F* [1, 9] = 49.075, *p* < 0.01 and *F* [4, 36] = 10.778, *p* < 0.01) and non-responders (*F* [1, 7] = 53.766, *p* < 0.01 and *F* [4, 28] = 0.355, *p* = 0.852) were observed. In addition, the post hoc comparisons revealed significant differences in the values at post0, post15, and post30, compared to those at baseline and the “between” time point in the WI priming trials in responders (*p* < 0.05, Bonferroni-corrected), and even in the values at post15 and post30, compared to those at baseline and the “between” time point in WI priming trials in non-responders (*p* < 0.05, Bonferroni-corrected). Moreover, there were significant differences in the values between the WI and CON trials at post30 in the responders (*p* < 0.05, Bonferroni-corrected) and at post15 and post30 in the non-responders (*p* < 0.05, Bonferroni-corrected). 

We tested whether the PAS25 response of each individual in the no priming trials was correlated with any of the baseline physiological measures collected. The PAS25 response in the CON trials was significantly related to RMT (*r* = −0.472, *p* < 0.05), AMT_pa_ (*r* = −0.582, *p* < 0.05) and TS_1mV_base_ (r = −0.473, *p* < 0.05), but not to the other baseline physiological measures (Table 2).

### 3.3. SAI and SICI 

Figure 5 shows the time courses of SICI_3ms_ (Figure 5A), SAI_N20+2_ (Figure 5B), SAI_N20+4_ (Figure 5C), and SAI_25ms_ (Figure 5D) in both the WI and CON trials. Two-way rmANOVA of the SICI_3ms_ revealed no interaction between “trial” and “time” (*F* [6, 102] = 0.796, *p* = 0.576), and no main effects of “trial” (*F* [1, 17] = 0.026, *p* = 0.874) and “time” (*F* [6, 102] = 0.687, *p* = 0.661) either. 

Two-way rmANOVA on the SAI_N20+2_ revealed a significant interaction between “trial” and “time” (*F* [6, 102] = 3.255, *p* < 0.01), as well as the main effects of “trial” (*F* [1, 17] = 1.251, *p* = 0.279) and “time” (*F* [6, 102] = 4.120, *p* < 0.01). Post hoc comparisons revealed significant differences in the values at post1, compared to those at baseline, as well as in the values at post0 and at post2, compared to those at baseline and post5 only in the WI trials (*p* < 0.05, Bonferroni-corrected); however, there were no significant pair-wise differences in the CON trials. Moreover, a significant difference was observed in the values between the CON and WI trials at post1 (*p* < 0.05, Bonferroni-corrected).

Two-way rmANOVA of the SAI_N20+4_ revealed a significant interaction between “trial” and “time” (*F* [6, 102] = 2.898, *p* < 0.05), as well as the main effects of “trial” (*F* [1, 17] = 4.938, *p* < 0.05) and “time” (*F* [6, 102] = 5.606, *p* < 0.01). Post hoc comparisons revealed significant differences in the values at post1, compared to those at baseline and post0, as well as in the values at post2, compared to those at baseline, post0, post3, post4, and post5 only in the WI trials (*p* < 0.05, Bonferroni-corrected). Alternatively, there were no significant pair-wise differences in the values in the CON trials. Nevertheless, significant differences were found in the values between the CON and WI trials at post1 and post2 (*p* < 0.05, Bonferroni-corrected).

Two-way rmANOVA of the SAI_25ms_ revealed a significant interaction between “trial” and “time” (*F* [6, 102] = 2.307, *p* < 0.05), as well as the main effect of “trial” (*F* [1, 17] = 6.200, *p* < 0.05), but no main effect of “time” (*F* [6, 102] = 1.714, *p* = 0.125). Moreover, post hoc comparisons revealed significant differences in the values at post1 and post2, compared to those at baseline only in the WI trials (*p* < 0.05, Bonferroni-corrected); however, there were no significant differences in the values in the CON trials. In addition, there were significant differences in the values between the CON and WI trials at post1 and post2 (*p* < 0.05, Bonferroni-corrected).

The ICC values calculated to test the consistency between the inhibitory values measured in the same individual on different days were 0.881 for SICI_3ms_, 0.893 for SAI_N20+2_, 0.924 for SAI_N20+4_, and 0.969 for SAI_25ms_ (all *p* < 0.001) (Figure 6). We tested whether the PAS25 responses in each individual obtained in experiment 1 were correlated with any of the inhibitory measures collected in experiment 2, and which found a significant correlation between the PAS25 response and SAI_25ms_ (*r* = −0.534, *p* < 0.05) (Figure 7, Table 2). 

### 3.4. MEP Recruitment Curve

Figure 8 illustrates the MEP recruitment curves before, during, and after the WI intervention. Two-way rmANOVA revealed no interaction between “intensity” and “time” (*F* [3.67, 29.33] = 1.08, *p* = 0.36), and no main effect of “time” (*F* [2.40, 19.16] = 0.89, *p* = 0.45), although there was a main effect of “intensity” (*F* [1.14, 9.12] = 32.94, *p* < 0.00).

## 4. Discussion

We present two major findings. First, SAI with an ISI of 25 ms appeared to be predictive of PAS25-induced plasticity, whereas I-wave recruitment did not. Second, the PAS25 response was enhanced by prior WI. This priming effect was observed in 17 participants, irrespective of whether they were responders or non-responders to PAS25 alone, as there was little inter-individual variability in the homeostatic SAI increase after WI. These results support our initial hypothesis that WI priming enhances the PAS25-induced plasticity in the primary motor cortex. 

WI can alter numerous physiological parameters and has recently been used to improve the activities of daily living [40,41] and cognitive function in the elderly subjects [42]. Moreover, WI can influence neural activity, such as activation in the primary somatosensory area and multimodal sensory processing [21]. However, to our knowledge, no study had examined the WI priming effect on neural plasticity, despite the observed decrease in cholinergic activity during WI [39]. Priming the activity of specific interneurons synapsing in M1 was found to alter the neural plasticity without modifying M1 excitability [43,44], thus supporting the notion that heterosynaptic plasticity may account for the effects of priming. Experiments 2 and 3 indicated that WI did not alter corticospinal excitability or intracortical excitability, consistent with previous studies [39,45]. In contrast, the cholinergic activity, as evaluated by SAI, significantly increased for at least 8.5 min after WI and returned to baseline at 21.5 min post-WI. Thus, the results of experiment 1 suggest that this cholinergic facilitation phase after WI may enhance the M1 plasticity induced by PAS25. 

Consistent with previous studies examining synaptic plasticity in the human motor cortex induced by TMS, we observed a large inter-individual variability in the PAS25 response. In particular, there was no significant response to PAS25 in the entire cohort, and only 55% of participants exhibited MEP facilitation after PAS25, which is within the proportion range noted in previous studies (e.g., 22% [11], 64% [10], and 68% [8]). This inter-individual variability was associated with RMT and AMT_pa_, but no other initial thresholds, including the indices of I-wave recruitment. The effect of attention [28,46,47] was unlikely because the participants were instructed to focus on the median nerve stimuli during the PAS25 protocol. Circadian effects [29] were also unlikely because all the experiments were performed in the afternoon. Finally, prior and ongoing voluntary muscle contractions [8,48] were carefully monitored and corrupted trials were exempted from the analysis. Therefore, the inter-individual variability presumably reflected the intrinsic differences in the PAS25 response between participants. 

Furthermore, responders exhibited lower RMT, AMT_pa_, and TS_1mV_base_ than non-responders. Muller-Dahlhaus et al. [9] examined the age-dependence of the PAS25 response and found that RMT and TS_1mV_base_ were predictive of PAS25 facilitation in elderly. However, the present study enrolled only young adults. Another possible explanation for the lower thresholds in responders is genetic polymorphisms associated with synaptic plasticity. For example, a single-nucleotide missense polymorphism in the gene encoding brain-derived neurotrophic factor (BDNF Val66Met) modified the experience-dependent motor cortical plasticity [49] and the PAS25 response [50]. This notion is also supported by our findings that responders showed greater M1 excitability than non-responders, and that the threshold correlated negatively with the PAS25 response. Motor thresholds and TS_1mV_base_ may thus reflect a genetically determined endophenotype [51] determining the magnitude and direction of stimulation-induced M1 plasticity in a given individual. 

We found no evidence that I-wave recruitment is related to PAS25 response, although a relationship between intermittent theta burst stimulation (iTBS)-induced neural plasticity and I-wave recruitment was reported by using the same methods [13]. A previous study showed that PAS25 led to a pronounced increase in the excitability of the cortical circuits generating later I-waves, whereas the earliest I-wave remained unaffected [52]. We hypothesized that the individual variability in the PAS25 response depends on the relative recruitment of late versus early I-waves. However, our results show that the recruitment of later I-waves is unlikely to predict the PAS25 response. This may be attributed to the distinct mechanisms for PAS25- and iTBS-evoked plasticity. PAS25 is a form of spike timing-dependent plasticity (STDP) [53], whereas TBS responses are related to activity-dependent changes in the synaptic strength between cortical neurons [54]. Moreover, the GABAergic activity involved in SICI was not affected by PAS25, but was decreased after iTBS [55]. 

The inter-individual variability in the PAS25 response was significantly related to SAI_25ms_, although it was not correlated with SICI_3ms_, SAI_N20+2_, or SAI_N20+4_. Thus, a higher SAI induced by a longer ISI (25 ms) predicted a greater PAS25 response, possibly because of an optimal interval between the pairing of conditioning afferent stimulation with TMS stimulation. SAI is induced by the increased excitability of GABAergic neurons through the activation of thalamocortical cholinergic projections [56]. The PAS25 induction protocol consists of repeated stimulus pairings with an ISI of 25 ms every 5 s, which increases the late I-wave amplitude without changing the I1 wave [52]. This LTP-like phenomena is reminiscent of STDP and may be produced either by a change in the efficacy of excitatory synaptic connections between the thalamocortical cholinergic projections activated by peripheral nerve stimulation or between P2−P3 pyramidal neurons activated by TMS, or by a change in the excitability of inhibitory neurons directly activated by afferent inputs and projecting onto P5 cells [56]. Considering these shared mechanisms, the present correlation between PAS25 and SAI_25ms_ appears reasonable. Strigaro et al. [57] showed that repeated pairing stimulation with the same ISI is important for the induction of the PAS plasticity response. Unfortunately, we cannot conclude that SAI_25ms_ is a predictor of the PAS25 response due to the lack of data showing whether the PAS changes at other ISIs (N20 + 2 and N20 + 4) can be predicted by SAI responses to the same ISI. 

Priming WI enhanced the PAS25 response in almost all participants, although there was no significant MEP facilitation by PAS25 alone due to the high inter-individual variability. Metaplasticity could explain these results. Animal experiments have shown that prior synaptic activity can influence the subsequent response to a stimulation protocol designed to induce LTP or LTD [4], and priming effects have been investigated in humans using non-invasive brain stimulation [18,19,48,58]. The LTP-like response to a facilitatory PAS protocol was decreased when preceded by similar facilitatory PAS, but enhanced when preceded by a PAS protocol that alone induced a LTD-like MEP decrease [19]. These results are consistent with the influence of homeostatic–homosynaptic metaplasticity. However, considering that RMT, MEP size, and SICI did not change 30 min after WI in the present study, the occurrence of metaplasticity among M1 pyramidal neurons and GABAAergic interneurons is less likely than a change in the cholinergic activity. 

The significantly increased SAI in experiment 2 indicated the facilitation of cholinergic activity between sensory and motor areas after WI. WI reportedly decreases SAI through widespread somatosensory inputs from the entire body surface [39]. Therefore, we speculate that a homeostatic response of cholinergic neurons between the sensory and motor areas occurs after WI, which could facilitate the PAS25 response. Cholinergic activity is a powerful regulator of synaptic plasticity. In animal studies, cholinergic blockade reduced LTP in the hippocampus, piriform cortex, and neocortex [59,60,61,62,63]; in contrast, in humans, use-dependent plasticity in the motor cortex was facilitated by an acetylcholinesterase inhibitor and blocked by a cholinergic antagonist [64,65]. The regulation of plasticity by cholinergic activity may facilitate the detection of incoming afferent inputs and decrease intrinsic feedback excitability, thus enhancing the encoding of relevant associated information [66,67]. Kuo et al. [24] reported that acetylcholine (Ach) enhanced the synapse-specific cortical excitability increase induced by PAS25 and consolidated the PAS10-induced reduction in motor cortical excitability, whereas it prevented the global excitatory aftereffects produced by anodal tDCS. They described that cholinergic nervous activity improved the efficacy of PAS by enhancing the signal-to-noise ratio, thereby facilitating the processing of meaningful (associative) inputs, and by suppressing non-meaningful/irrelevant asynchronous inputs [68,69,70]. Therefore, the enhanced plasticity following WI may be related to increased cholinergic activity associated with WI-induced homeostatic aftereffects. 

In the field of neurological rehabilitation, not all patients can achieve expected results by multiple rehabilitation rounds due to several reasons. However, it should be noted that M1 plasticity plays essential roles in motor learning and memory, which are integral to neurological rehabilitation. Similar to results in previous studies [10,71], the present results show a high inter-individual variability for PAS25 and M1 plasticity. On the basis of these, the present finding that WI priming will promote LTP-like plasticity induction may apply to home-based rehabilitation. WI prior to starting rehabilitation will promote M1 plasticity induction, and this may help to facilitate the effects of rehabilitation. To date, beyond this theory, no direct evidence has been found to show that WI priming facilitates the effects of rehabilitation, hence, further studies are required in this direction.

The present results suggest that WI prior to PAS25 facilitates LTP-like plasticity due to a homeostatic increase in the cholinergic activity. The I-wave recruitment evaluated by the MEP latency difference was not related to the PAS25 plasticity response, in contrast to the TBS response [13]. 

## Figures and Tables

**Figure 1 ijerph-17-00215-f001:**
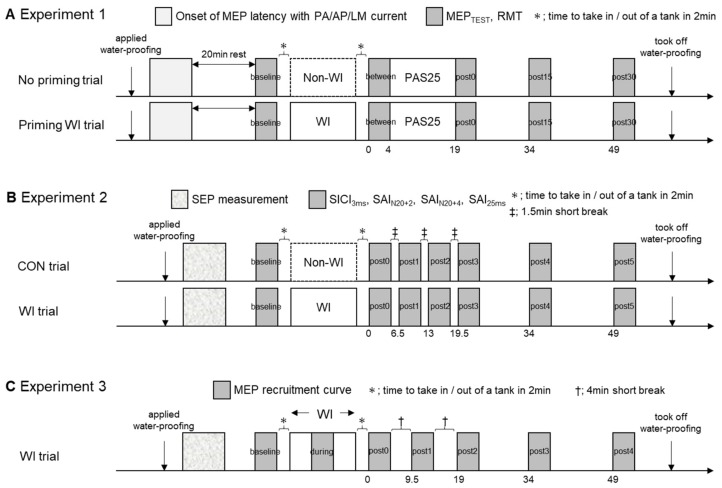
The experimental protocol. (**A**) Experiment 1 examined the priming effect of water immersion (WI) on paired associative stimulation with ISI of 25 ms (PAS25) by comparison between no priming and priming WI trials. After measuring onset of motor-evoked potential (MEP) latency induced by single pulse transcranial magnetic stimulation (TMS) with posterior-to-anterior (PA), anterior-to-posterior (AP), and lateral-to-medial (LM) current during a little contraction, MEP_TEST_ and resting motor threshold (RMT) were measured at baseline, before PAS25 (between), and at 0 min, 15 min, and 30 min after PAS25 (post0, post15, and post30). (**B**) Experiment 2 examined the effects of WI on short-interval intracortical inhibition (SICI) and short-latency afferent inhibitions (SAIs) by comparison between non-WI (CON) and WI trials. After short latency evoked potential (SEP) measurement to determine stimulus paradigm of SAIs, SICI and SAIs were measured before (baseline) and after intervention (post0, post1, post2, post3, post4, and post5). (**C**) Experiment 3 investigated the changes in MEP recruitment curve before (baseline), during (during), and after WI (post0, post1, post2, post3, and post4).

**Figure 2 ijerph-17-00215-f002:**
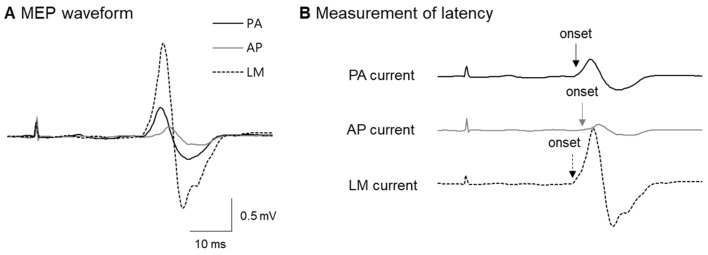
Representative MEP waveform and measurement of MEP latency. (**A**) Schematic representation of the coil orientations and typical examples of MEPs during contraction by each stimulus. (**B**) Arrow indicates the timing of TMS and arrowhead indicates the onset of MEPs. The PA–LM latency difference was 1.3 ms in this case, whereas AP–LM was 3 ms, compatible with known latency differences between D- and I1-waves or I3-waves [32].

**Figure 3 ijerph-17-00215-f003:**
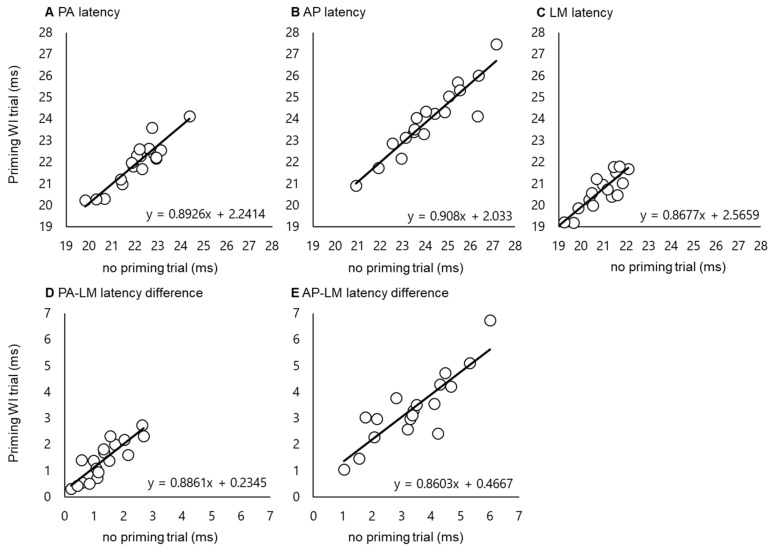
Test-retest reliability of each MEP latency (**A**–**C**) and latency difference (**D**,**E**). Black lines show regression lines in each parameter. The *x*- and *y*-axis show each value in no priming and priming WI trials. Intraclass correlation coefficient (ICC) values (detail in results) mean higher test–retest reliability in all parameters.

**Figure 4 ijerph-17-00215-f004:**
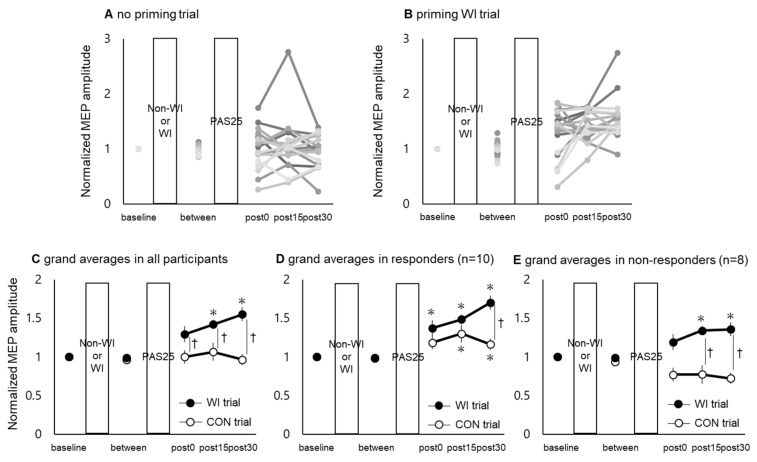
Time course of PAS25. (**A**,**B**) show individual responses to PAS25 in no priming and priming WI trials, respectively. Though there was high inter-individual variability for PAS25 response, (**C**) shows significant group difference of PAS25 response—increased in WI trial and unchanged in CON trial. (**D**,**E**) show PAS25 response in responders and non-responders. In responders, PAS25 response was facilitated in WI trial compared to CON trial. In non-responders, expected PAS25 responses were present in WI trial, whereas there were opposite responses or no response for PAS25 in CON trial. The *x*- and *y*-axes show measurement time points and MEP amplitude normalized to baseline, respectively. Asterisk (*) represents significant difference compared to baseline and “between” time points (*p* < 0.05). Dagger (†) shows significant difference between no priming and priming WI trials (*p* < 0.05).

**Figure 5 ijerph-17-00215-f005:**
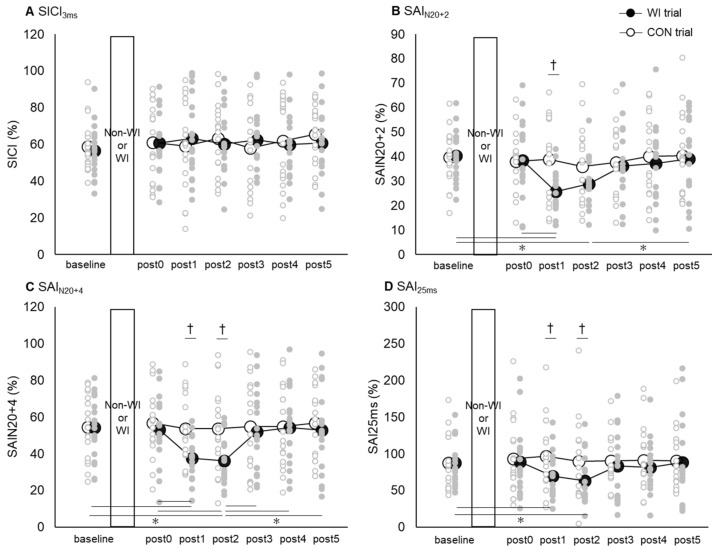
Time course of SICI_3ms_ (**A**), SAI_N20+2_ (**B**), SAI_N20+4_ (**C**), and SAI_25ms_ (**D**) in both WI and CON trials. (**A**) shows the change in SICI_3ms_ before and after WI, which, as shown, did not change throughout the experiment. (**B**–**D**) present the change in SAI_N20+2_, SAI_N20+4_, and SAI_25ms_ before and after WI, which all showed temporal increase in SAI after WI irrespective of ISI. The *x*- and *y*-axes show measurement time points and each value, respectively. Asterisk (*) represents significant difference among each time points (*p* < 0.05). Dagger (†) shows significant difference between non-WI and WI trials (*p* < 0.05).

**Figure 6 ijerph-17-00215-f006:**
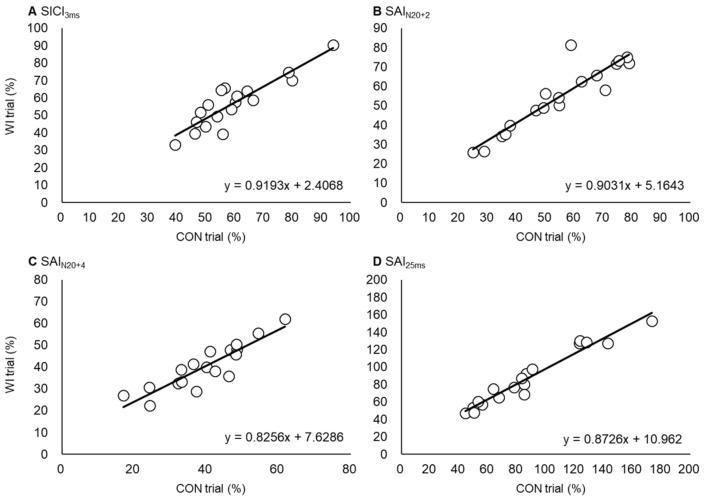
Test–retest reliability of SICI_3ms_ (**A**), SAI_N20+2_ (**B**), SAI_N20+4_ (**C**), and SAI_25ms_ (**D**). Black lines show regression lines in each parameter. The *x*- and *y*-axes show each value in CON and WI trials. ICC values (detail in results) mean higher test–retest reliability in all parameters.

**Figure 7 ijerph-17-00215-f007:**
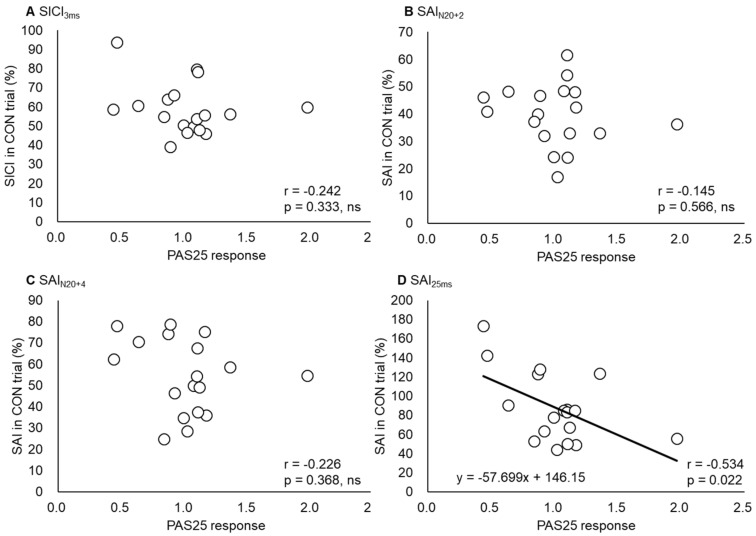
The relationships between PAS25 response and SICI_3ms_ (**A**), SAI_N20+2_ (**B**), SAI_N20+4_ (**C**), and SAI_25ms_ (**D**). Black lines show regression lines in each parameter. The *x*- and *y*-axes show PAS25 response and each value in CON trials, respectively. As shown in (**A**–**C**), there was no correlation between PAS25 response and each value (SICI_3ms_, SAI_N20+2_, SAI_N20+4_). On the other hand, (**D**) shows a significant negative correlation between PAS25 response and SAI_25ms_, which indicates that SAI_25ms_ would be a predictor for PAS25 response.

**Figure 8 ijerph-17-00215-f008:**
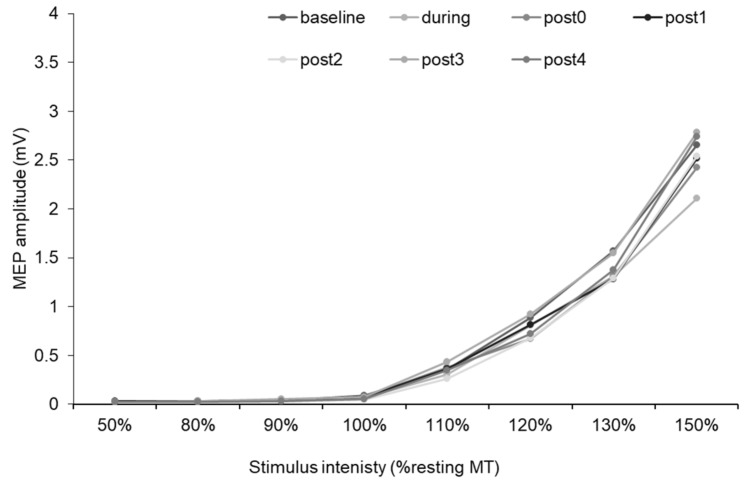
MEP recruitment curves before, during, and after WI. The *x*- and *y*-axes show stimulus intensity of TMS (% resting MT) and MEP amplitude, respectively. MEP recruitment curves were measured before, during, and after WI. There was no change in MEP recruitment curve, which indicates that WI did not change corticospinal excitability.

**Table 1 ijerph-17-00215-t001:** Participant characteristics and parameters of TMS and ES in experiments 1 to 3.

Experiment 1		
Participants’ Characteristics		
Number	18 (female 3, male 15)	
Age (years old)	21.37 ± 0.18	
TMS and ES Parameter		
	No Priming	Priming WI
RMT (%)	42.28 ± 1.93	40.72 ± 1.68
AMT_pa_ (%)	34.83 ± 1.55	32.61 ± 1.29
AMT_ap_ (%)	49.89 ± 2.24	47.50 ± 2.00
AMT_lm_ (%)	37.56 ± 1.67	36.50 ± 1.26
TS_1mV_base_	52.72 ± 2.37	52.83 ± 2.23
ST (mA)	3.63 ± 0.28	3.73 ± 0.34
MEP Latency		
PA (ms)	22.10 ± 0.26	21.96 ± 0.25
AP (ms)	24.17 ± 0.39	23.98 ± 0.38
LM (ms)	20.77 ± 0.26	20.59 ± 0.25
Latency Differences		
PA–LM (ms)	1.33 ± 0.17	1.38 ± 0.19
AP–LM (ms)	3.40 ± 0.32	3.39 ± 0.31
AP–PA (ms)	2.08 ± 0.25	2.02 ± 0.28
**Experiment 2**		
**Participants’ Characteristics**		
Number	18 (female 3, male 15)	
Age (years old)	21.37 ± 0.18	
N20 (msec)	18.62 ± 0.16	
TMS and ES Parameter		
	CON Trial	WI Trial
RMT (%)	45.29 ± 1.65	44.00 ± 1.02
AMT_pa_ (%)	35.12 ± 1.26	33.61 ± 0.83
TS_1mV_base_	58.88 ± 1.99	57.44 ± 1.92
ST (mA)	3.35 ± 0.21	3.67 ± 0.20
CS_SICI_ (%)	31.46 ± 1.22	30.48 ± 0.71
CS_SAI_ (mA)	10.04 ± 0.64	10.97 ± 0.55
**Experiment 3**		
**Participants’ Characteristics**		
Number	9 (female 1, male 8)	
Age (y.o.)	21.89 ± 0.31	
RMT (%)	42.71 ± 1.77	

ES: electrical stimulation; AMT: active motor threshold; TS: test stimulus; PA: posterior-to-anterior; AP: anterior-to-posterior; LM: lateral-to-medial; ST: sensory threshold; CS: conditioning stiumulus.

**Table 2 ijerph-17-00215-t002:** Relationship between PAS25 response and each baseline value in experiments 1 and 2.

Experiment 1		
	*r*	*p*
RMT	−0.472	0.048 *
AMT_pa_	−0.582	0.011 *
AMT_ap_	−0.351	0.154
AMT_lm_	−0.348	0.157
TS_1mV_base_	−0.437	0.047 *
ST	0.288	0.246
PA latency	0.169	0.502
AP latency	0.040	0.876
LM latency	0.095	0.706
PA–LM	0.127	0.615
AP–LM	−0.019	0.940
**Experiment 2**		
	***r***	***p***
SICI_3ms_	−0.24	0.333
SAI_N20+2_	−0.15	0.566
SAI_N20+4_	−0.23	0.368
SAI_25ms_	−0.53	0.022 *

* Significant correlation with PAS25 response.

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
