# Peer review of "Priming Effects of Water Immersion on Paired Associative Stimulation-Induced Neural Plasticity in the Primary Motor Cortex"

_ijerph, 2019, doi:10.3390/ijerph17010215_

Round 1
Reviewer 1 Report
The main aim of the study was to verify whether water immersion (WI) can facilitate the PAS25-induced plasticity after having tested whether indirect-wave (I-wave) recruitment and cortical inhibition can predict the response to paired associative stimulation with an inter-stimulus interval of 25 ms (PAS25). The authors used a stimulation protocol allowing to evaluate the short-latency afferent inhibition (SAI) and the short-interval intracortical inhibition (SICI). Importantly, the authors examined the motor evoked potential (MEP) amplitudes before and after PAS25, with and without WI, and used the SAI, SICI, and MEP recruitment curves to determine the mechanism underlying priming by WI on PAS25. Results showed that SAI, with an inter-stimulus interval of 25 ms, might serve as a predictor of the response to PAS25, whereas I-wave recruitment evaluated by the MEP latency difference was not predictive of the PAS25 response. Relevantly, results also showed that 15-min WI prior to PAS25 facilitated LTP-like plasticity.
The study is well-conducted, the methodology is sound, and the results are clear. In particular, the authors compared responders and non-responders trying to clarify recent data showing that PAS25-induced plasticity is not observed in 25−64% of healthy young volunteers. Since the history of prior cortical activity may be among possible factor influencing the PAS25 response, this factor was also manipulated by WI.
I have one main point, that is not a minor one, relating to the translational aspects of the results.
In the last paragraph of the Introduction section, the authors underline that elucidation of the effects of WI priming on the neural plasticity of M1 “could…. facilitate the development of improved aquatic therapies for neurological patients”.
However, although the authors suggest that their result can support the development of improved aquatic therapies for neurological patients, they do not provide any comment about this point Discussion section. Taking into account the main aims of the Journal, I do believe that this is not a secondary issue but rather a relevant one. The authors should Discuss this aspect explicating the way in which their results translate into applied contexts.
Author Response
Reviewer 1
The main aim of the study was to verify whether water immersion (WI) can facilitate the PAS25-induced plasticity after having tested whether indirect-wave (I-wave) recruitment and cortical inhibition can predict the response to paired associative stimulation with an inter-stimulus interval of 25 ms (PAS25). The authors used a stimulation protocol allowing to evaluate the short-latency afferent inhibition (SAI) and the short-interval intracortical inhibition (SICI). Importantly, the authors examined the motor evoked potential (MEP) amplitudes before and after PAS25, with and without WI, and used the SAI, SICI, and MEP recruitment curves to determine the mechanism underlying priming by WI on PAS25. Results showed that SAI, with an inter-stimulus interval of 25 ms, might serve as a predictor of the response to PAS25, whereas I-wave recruitment evaluated by the MEP latency difference was not predictive of the PAS25 response. Relevantly, results also showed that 15-min WI prior to PAS25 facilitated LTP-like plasticity.
The study is well-conducted, the methodology is sound, and the results are clear. In particular, the authors compared responders and non-responders trying to clarify recent data showing that PAS25-induced plasticity is not observed in 25−64% of healthy young volunteers. Since the history of prior cortical activity may be among possible factor influencing the PAS25 response, this factor was also manipulated by WI.
I have one main point, that is not a minor one, relating to the translational aspects of the results.
Reply:
We thank the reviewer for the helpful and insightful comments that have helped us to improve our manuscript. We are glad to hear from you that our study was well conducted. We have listed our point-by-point responses to your comments below.
In the last paragraph of the Introduction section, the authors underline that elucidation of the effects of WI priming on the neural plasticity of M1 “could…. facilitate the development of improved aquatic therapies for neurological patients”.
However, although the authors suggest that their result can support the development of improved aquatic therapies for neurological patients, they do not provide any comment about this point Discussion section. Taking into account the main aims of the Journal, I do believe that this is not a secondary issue but rather a relevant one. The authors should Discuss this aspect explicating the way in which their results translate into applied contexts.
Reply:
We thank you for suggesting that this study should be expanded along the context of application in the development of improved aquatic therapies. As the reviewer pointed out, we had no discussion and explanation in the original draft for the application of the present results in that regards. Consequently, we have added the following paragraph to the final part of discussion section as shown below:
[Insert, page 29, line 22, revised draft]
In the field of neurological rehabilitation, not all patients can achieve expected results by multiple rehabilitation rounds due to several reasons. However, it should be noted that M1 plasticity plays essential roles in motor learning and memory, which are integral to neurological rehabilitation. Similar to results in previous studies [10, 71], the present results show a high inter-individual variability for PAS25 and M1 plasticity. Based on these, the present finding that WI priming will promote LTP-like plasticity induction may apply to home-based rehabilitation. WI prior to starting rehabilitation will promote M1 plasticity induction, and this may help to facilitate the effects of rehabilitation. To date, beyond this theory, no direct evidence has been found to show that WI priming facilitates the effects of rehabilitation, hence, further studies are required this direction.
Reviewer 2 Report
The current study had two major purposes which were to determine if I wave recruitment and cortical inhibition can regulate or predict PAS responses and to determine if water immersion could facilitate PAS induced plasticity. A total of 3 experiments were performed with 18, 18, and 9 subjects in experiments 1, 2, 3, respectively. Numerous TMS measures (APB muscle) were conducted including MEPS and their latencies induced by PA, LM, and AP directed currents, paired pulse measures (SAI, SICI), and PAS induced changes in cortical excitability. In addition, MEP recruitment curves, RMT, 1 mV MEPs, and AMT were quantified when appropriate and to obtain the parameters for the paired pulse measures. Briefly, the main findings were: 1) SAI at 25 ms was predictive PAS25 induced plasticity, although I-wave recruitment was not; and 2) PAS25 was enhanced by water immersion. The authors imply that water immersion could be a potential therapeutic intervention to modify some cortical pathways, although this seems to be somewhat of a stretch. Nonetheless, the authors did not dwell or overspeculate on this point.
Overall, the manuscript is very well-written, appears to have few grammatical or typographical errors, and extends previous research. The study design and analyses appear to appropriate and the data appear to have been collected carefully. The technical difficulty of the experiments was high and the description of the methodology was very good. In addition, the authors also did a good job of explaining the results of their study in light of previous findings. The topic addressed is relatively novel (I was not aware of the previous research on cortical responses to water immersion). The focus of the research seems to be appropriate for the journal and of interest to many readers of the journal. There appear to be no major flaws in the paper and I do not have major concerns with the study design and interpretation of results. I only have a few minor points (see below).
Minor Points:
Page 4 lines 7-8. Is it correct that metaplasticity has been described as a neuroprotective mechanism? Please check that this is correct I have not seen it described that way.
Page 7, Line 18. Why was the counting done? Page 7, Lines 21 -27 and the whole paragraph. Why were these water immersion parameters chosen? I assume based on previous research by the group? Please clarify. Figure 6. I think the X axis should say % instead of @. Please fix this typo.Author Response
Reviewer 2
The current study had two major purposes which were to determine if I wave recruitment and cortical inhibition can regulate or predict PAS responses and to determine if water immersion could facilitate PAS induced plasticity. A total of 3 experiments were performed with 18, 18, and 9 subjects in experiments 1, 2, 3, respectively. Numerous TMS measures (APB muscle) were conducted including MEPS and their latencies induced by PA, LM, and AP directed currents, paired pulse measures (SAI, SICI), and PAS induced changes in cortical excitability. In addition, MEP recruitment curves, RMT, 1 mV MEPs, and AMT were quantified when appropriate and to obtain the parameters for the paired pulse measures. Briefly, the main findings were: 1) SAI at 25 ms was predictive PAS25 induced plasticity, although I-wave recruitment was not; and 2) PAS25 was enhanced by water immersion. The authors imply that water immersion could be a potential therapeutic intervention to modify some cortical pathways, although this seems to be somewhat of a stretch. Nonetheless, the authors did not dwell or overspeculate on this point.
Overall, the manuscript is very well-written, appears to have few grammatical or typographical errors, and extends previous research. The study design and analyses appear to appropriate and the data appear to have been collected carefully. The technical difficulty of the experiments was high and the description of the methodology was very good. In addition, the authors also did a good job of explaining the results of their study in light of previous findings. The topic addressed is relatively novel (I was not aware of the previous research on cortical responses to water immersion). The focus of the research seems to be appropriate for the journal and of interest to many readers of the journal. There appear to be no major flaws in the paper and I do not have major concerns with the study design and interpretation of results. I only have a few minor points (see below).
Reply:
We are glad to hear from you that our manuscript was well written. We have listed our point-by-point responses to your comments below.
Minor Points:
Page 4 lines 7-8. Is it correct that metaplasticity has been described as a neuroprotective mechanism? Please check that this is correct I have not seen it described that way.
Reply:
Thank you for your important question. The review article by Müller-Dahlhaus and Zeimann (2015) described that metaplasticity has a neuroprotective effect that stabilizes synaptic weights in neuronal networks while maintaining the capacity for synaptic plasticity. Specifically, homeostatic metaplasticity, as formalized in the Bienenstock–Cooper–Munro (BCM) theory of bidirectional synaptic plasticity (Bienenstock and others 1982), states that “the synaptic modification threshold θM, that is, the threshold for induction of LTP versus LTD, is not stable but varies as a function of the integrated postsynaptic activity: it decreases at low levels of previous postsynaptic activity, favoring induction of LTP over LTD.” “Conversely, θM increases at high levels of recent postsynaptic activity, thereby favoring the probability of subsequent LTD over LTP. This sliding synaptic modification threshold thus enables maintenance of neuronal and network activity in a physiological range.”
Page 7, Line 18. Why was the counting done?
Reply:
Thank you for your question. A previous study had reported that attentional control influenced plasticity induction by PAS25 (Stefan K et al. 2004). Therefore, we instructed participants to count the number of stimuli to maintain participants’ attention.
Page 7, Lines 21 -27 and the whole paragraph. Why were these water immersion parameters chosen? I assume based on previous research by the group? Please clarify.
Reply:
As you guessed, we referenced previous studies, which have reported WI induced change in SAI as an evaluation of cholinergic neural activity, and the subsequent autonomic nervous activity as an indicator of arousal level (Sato D et al. 2013; Sato D et al. 2017).
Figure 6. I think the X axis should say % instead of @. Please fix this typo.
Reply:
Thank you for your comment. We have changed “@” to “%.”
Reviewer 3 Report
The manuscript is a very interesting and well-designed work. Results are in general well described and cleverly analysed. Nevertheless, I have some minor questions with the aim to improve the manuscript.
1.- The running title is practically the same than the main title. Please, modify.
2.- Pag. 2. How do you measure MEP, to the first negative peak, from peak to peak?. Please, discuss whether measurement of area, instead of voltage for MEP could modify the results.
3.- Pag. 2. How do you measure a 10% of contraction of a muscle?. Explain.
4.- Legends and labels in figures are not legible. It is be very important to improve the quality of figures.
5.- .It should be important to show (maybe in Appendices, I don’t know) a raw recording of MEP and I waves, explaining measurements: amplitudes, latencies, etc.
6.- The time unit at the International System, the second is denoted as s, not sec.
7.- I don’t find clearly how were non-responders defined. Please, explain.
8.- Figure 2. Labels A and B are cited in the opposite at main text (pag 17).
9.- All the legends must be much better described.
Author Response
Reviewer 3
The manuscript is a very interesting and well-designed work. Results are in general well described and cleverly analysed. Nevertheless, I have some minor questions with the aim to improve the manuscript.
Reply:
We thank the reviewer for the helpful and insightful comments that have helped us to improve our manuscript. We are glad to hear from you that our study was well designed. We have listed our point-by-point responses to your comments below.
1.- The running title is practically the same than the main title. Please, modify.
Reply:
We have changed the running title to “WI priming effects on M1 plasticity .”
2.- Pag. 2. How do you measure MEP, to the first negative peak, from peak to peak?. Please, discuss whether measurement of area, instead of voltage for MEP could modify the results.
Reply:
Thank you for your question about how we measured MEP amplitude. We measured MEP amplitude from peak-to-peak burrowing a leaf from numerous previous studies, which reported that M1 plasticity changes by non-invasive brain stimulation including PAS25, rTMS and tDCS (Di Lazzaro V et al. 2005; Rosenkranz K et al. 2007; Hamada M et al. 2008; Ridding MC and U Ziemann 2010; Di Lazzaro V et al. 2018).
3.- Pag. 2. How do you measure a 10% of contraction of a muscle?. Explain.
Reply:
As the reviewer has pointed out, it is very important to measure the correct response for TMS during contraction. Regarding this, we have added the following to the methodology in the revised draft:
[Insert, page 6, line 21, revised draft]
To maintain 10% contraction, as performed by previous studies [26], a rectified running average EMG with an averaging window of 175 ms was used to provide visual feedback on the monitor to the participants.
4.- Legends and labels in figures are not legible. It is be very important to improve the quality of figures.
Reply:
Thank you. We have modified all figures and legends and they are now legible.
5.- .It should be important to show (maybe in Appendices, I don’t know) a raw recording of MEP and I waves, explaining measurements: amplitudes, latencies, etc.
Reply:
We thank you for your suggestion. We have added Figure 2, which explains the raw recording of MEP and I-wave recruitment in the methods section.
6.- The time unit at the International System, the second is denoted as s, not sec.
Reply:
Thank you. We have changed “msec” to “ms.”
7.- I don’t find clearly how were non-responders defined. Please, explain.
Reply:
Thank you. We defined the responder and non-responder groups according to the presence of grand average PAS25 responses of below and above 1. We have written this in the methods section and you can find it in page 14, line 15.
8.- Figure 2. Labels A and B are cited in the opposite at main text (pag 17).
Reply:
Thank you for your observation. We have resolved the issue.
9.- All the legends must be much better described.
Reply:
Thank you for your suggestion. We have improved upon the description of all the legends.